# The Effectiveness of the Addition of Platelet-Rich Fibrin to Bovine Xenografts in Sinus and Bone Ridge Augmentation: A Systematic Review

**DOI:** 10.3390/jfb14070389

**Published:** 2023-07-23

**Authors:** Katia Idiri, Octave Bandiaky, Assem Soueidan, Christian Verner, Emmanuelle Renard, Xavier Struillou

**Affiliations:** 1Faculté de Chirurgie Dentaire, CHU Nantes, Service Odontologie Restauratrice et Chirurgicale, Nantes Université, 44035 Nantes, France; katia.idiri@univ-nantes.fr (K.I.); octave.bandiaky@univ-nantes.fr (O.B.); assem.soueidan@univ-nantes.fr (A.S.); christian.verner@univ-nantes.fr (C.V.); emmanuelle.renard@univ-nantes.fr (E.R.); 2Oniris, CHU Nantes, INSERM, Regenerative Medicine and Skeleton, RMeS, Nantes Université, UMR 1229, F-44000 Nantes, France

**Keywords:** platelet-rich fibrin, BioOss, bone ridge augmentation, dental implantation

## Abstract

Dental implants sometimes need bone augmentation to recreate an adequate bone height and volume. Numerous bone augmentation techniques have been described, and, currently, the most commonly used bone graft procedure is xenografts with deproteinized bovine bone mineral (DBBM). The addition of platelet-rich fibrin (PRF) to DBBM has already shown better performance than DBBM alone in restoring intrabony periodontal defects, but the role of PRF in preimplantation bone grafts is still not clear. The objective of this systematic review was to evaluate the efficacy of the adjunction of PRF or L-PRF to DBBM in bone ridge augmentation procedures. Clinical randomized controlled studies using PRF associated with DBBM were included. In April 2023, three electronic databases (PubMed, Cochrane, and Web of Science) were searched. The search strategy was performed according to PRISMA guidelines. The risk of bias assessments were performed using the Cochrane Collaboration tool. A total of seven articles were included and analyzed. The results show no statistically significant effect of PRF added to DBBM compared to DBBM alone in the sinus lift procedure but do show an effect in the reduction in bone graft resorption in one study of mandibular guided bone regeneration.

## 1. Introduction

Implant dentistry is an established treatment involving the use of dental implants supporting restoration to replace missing teeth. It offers several advantages over other tooth replacement options, such as the preservation of adjacent teeth and the possibility of fixation [1]. However, missing teeth have consequences, including the resorption of alveolar bone, which induces horizontal and vertical dimension reduction [2]. The bone volume influences the operative protocol and the type of implant used to replace lost teeth. To treat alveolar ridge defects, many bone graft techniques have been developed over time to preserve or reconstruct the alveolar bone volume and enlarge implantation indications. The need for bone augmentation procedures appears to be a frequent requirement before implant placement, particularly in the maxillary or mandibular posterior region [3,4,5,6]. Autologous bone grafts represent the highest degree of biological safety but also have disadvantages, such as the requirement for a secondary surgical site, the need for general anesthesia, and high postoperative morbidity at the second site. The risks of scarring in the donor site, the tendency for partial resorption, and postoperative complications are also limitations in the use of autologous bone grafts [7,8]. Therefore, bone substitutes have been developed, such as xenogeneic or alloplastic materials. These materials must meet several essential criteria, including biocompatibility (the material does not induce adverse reactions or rejection), osteoconductivity (the materials must be able to support the growth of new bone tissue by providing a scaffold for bone regeneration), osteoinducibility (materials must be able to stimulate osteoblast arrival to regenerate new bone tissue), and resorbability (materials have to be gradually broken down and replaced by new bone tissue). Xenografts are grafting materials that are derived from a species genetically unrelated to the host, and the most common source of these materials in dental practice is deproteinized bovine bone mineral (DBBM). The most commonly used DBBM is the commercially available BioOss^TM^, produced by removing the organic components at high temperatures to minimize the immune response. This treatment results in a crystal structure that practically matches human cancellous bone in structure and provides biocompatible and osteoconductive properties [8]. The particle size of this material is 0.25 to 2 mm [9,10]. BioOss^TM^ has been shown to be effective in sinus floor augmentation [11], socket or alveolar ridge preservation, horizontal and vertical augmentation [12,13], and peri-implant defects [14,15].

To enhance wound healing and bone regeneration either after dental extraction or before or during implant placement, the use of bone substitutes associated with growth factors has been proposed based on the therapeutic concept that a supraphysiological concentration of growth factors better supports the early stages of wound healing and bone regeneration [16,17]. On this principle, the use of three-dimensional scaffolds produced from the patient’s own peripheral blood has been developed and proposed for preimplantation bone grafting. Platelet-rich fibrin (PRF) prepared from plasma after centrifugation of whole blood [18] contains platelets and leukocytes as well as a variety of growth factors and cytokines, including transforming growth factor-beta 1 (TGF-β1), platelet-derived growth factor (PDGF), vascular endothelial growth factor (VEGF), and interleukin (IL)-1β, IL-4, and IL-6 [19]. Plasma containing platelets and leukocytes undergoes spontaneous coagulation, such as in a natural blood clot, or can be further processed, resulting in a PRF membrane [19]. There are two forms of PRF: solid PRF and liquid PRF (L-PRF). The production of L-PRF is accomplished by using a lower centrifugation speed for a shorter duration than PRF. L-PRF can be mixed more effectively with particulate bone substitutes [20]. Numerous systematic reviews and meta-analyses have evaluated the effect of PRF in dental surgery in treatment of support periodontal soft tissue repair [21], periodontal intrabony defects [22,23], maxillary sinus floor augmentation [24,25,26,27], socket preservation [28], or all aspects of implant therapy [29]. These reviews show good effects, particularly for soft tissue and periodontal treatment, but do not yet show strong evidence of the benefits of PRF in bone regeneration. Although PRF is associated with biomaterials such as deproteinized bovine bone mineral (BioOss^TM^), it has shown better performance than BioOss^TM^ alone in restoring intrabony periodontal defects [30], and the effect of PRF combined with BioOss^TM^ on bone regeneration before implantation is still not clear. This systematic review aimed to evaluate the effect of PRF when mixed with BioOss^TM^ in comparison with the use of BioOss^TM^ alone in alveolar ridge augmentation and in the outcome of implantation after bone augmentation procedures.

## 2. Materials and Methods

The protocol for this review was registered in the Prospective International Registry of Systematic Reviews (PROSPERO) under ID CRD42023411160 and followed the 2020 preferred reporting items for systematic reviews and meta-analyses (PRISMA) guidelines [29]. The PICOS structure was used to define the inclusion criteria and to formulate the research question. Studies that met the following criteria were included:

Participants (P): adult patients with atrophic posterior maxilla and/or horizontal and/or vertical bone deficiency in the posterior region of the mandible who were candidates for implant therapy. Interventions (I): use of xenografts alone or combined with PRF in the bone ridge augmentation procedure. Comparison (C): comparisons of clinical, histological, histomorphometric, and radiological parameters were made between the test group (PRF and bovine bone xenograft) and the control (bovine bone xenograft alone). Outcome (O): the primary outcome was new bone formation evaluated with histomorphometric analysis. The secondary outcomes were other histomorphometric parameters, such as the percentage of residual bone and soft tissue, bone maturation, and X-ray outcomes. Study design (S): only randomized controlled trials (RCTs) were selected. Systematic reviews and meta-analyses, case reports, in vitro studies, in situ studies, letters to editors, and studies using bone substitutes other than PRF and bovine bone xenografts were excluded. The research question was as follows: Does the addition of PRF to bovine bone xenografts improve new bone formation compared to bovine bone xenografts alone in bone ridge augmentation procedures?

### 2.1. Search Strategy

The keywords used were selected from the HeTOP glossary of MeSH terms according to the scheme indicated in Table 1. In summary, an exhaustive search without language or publication date restrictions was performed until March 2023 in three databases (PubMed/Medline, Cochrane Library, and Web of Sciences) using the keywords. In addition, a manual search was performed on the references of eligible articles, and specialized journals were also consulted to identify other articles related to the research question. The documents from this extensive literature search were transferred to an EndNote^®^ library, and duplicates were removed.

### 2.2. Screening and Study Selection

The method used was a systematic review of the literature, and the research and article selection process were carried out independently by two authors (K.I. and X.S.). A calibration was performed to determine inter-examiner agreement in the study selection process. This calibration was performed according to the method described by Landis and Koch [31]. After achieving an appropriate level of agreement (*κ* ≥ 0.81), the reviewers (K.I. and X.S.) performed a methodical analysis of all study titles, abstracts, and full texts independently. Discrepancies were resolved, and consensus was achieved by engaging a third author (E.R.).

### 2.3. Article Selection and Data Extraction

The titles and abstracts of the obtained articles were screened based on determined eligibility criteria. The full texts of the remaining studies were assessed by the same authors. The data of the included studies when available were extracted by both reviewers (K.I. and X.S.) and verified and confirmed by two other authors (O.N.B. and E.R.). A table was previously established to provide support for collecting the following data: (1) name of first author, year, and country of publication; (2) study design and group characteristics; (3) number of patients and mean age; (4) study groups, PRF preparation technique, and type of bovine bone xenograft; (5) number of implants and dental implant brands; (6) follow-up implant survival and implant stability; (7) radiographical outcomes; (8) histomorphometric and histological outcomes; and (9) results.

### 2.4. Quality Assessment

The risk of bias assessment was performed with Cochrane’s Collaboration tool for assessing the risk of bias in randomized controlled trials [32]. Two authors (I.K. and E.R.) assessed and discussed the risk of bias in the included studies, and a third researcher (O.N.B.) was approached when necessary to resolve disagreements. The risk of bias level was determined to be low, unclear, or high according to the following criteria: (1) generation of the randomization sequence (selection bias), (2) concealment of the allocation (reporting bias), (3) blinding of the investigators and the participants (confusion bias), (4) blinding of the evaluation of the results (performance bias), (5) management of missing data (attrition bias), (6) selection of the reporter, and (7) other types of bias.

## 3. Results

### 3.1. Study Selection

The bibliographic search of all sources identified 457 articles (443 from databases, 2 from a previous version of the review, and 12 from other methods). Of these, 10 duplicate studies were removed using the reference manager EndNote^®^. A total of 400 articles were excluded after reading titles and/or abstracts, and 23 records were excluded since reports were not retrieved. A total of 24 full-text records were read and analyzed, and 17 records were excluded for reasons such as evaluation of another biomaterial than DBBM (*n* = 9), in vitro studies (*n* = 1), poor study design (*n* = 3), and systematic review (*n* = 4), as shown in the PRISMA flowchart in Figure 1. Seven records met the inclusion criteria and were included in the systematic review [33,34,35,36,37,38,39].

### 3.2. Description of Included Studies

The characteristics of the seven included articles are presented in Table 2. The number of subjects included varied from 7 to 60, with a general mean age of 51.20 years old. The study designs included split-mouth RCT (Irdem et al., 2021; Nizam et al., 2018; Tatullo et al., 2012) [34,36,37,38] and parallel RCT [33,35,39]. All studies used BioOss^TM^ for DBBM. The studies in [33,34,35,36,37] used L-PRF, and those in [38,39] used PRF, although the centrifugation speed and duration were similar to those of L-PRF production. The majority of the surgical procedures were sinus lifts for the treatment of maxillary sinus atrophy, except for one study where the patients were treated for guided bone regeneration in bone deficiency in the mandibular posterior region [35]. The implant number varied from 5 to 50 per group. Implant diameter and length were specified in two studies [35,36], and [36,39] specified implant marks. The maximum clinical and radiographic follow-up of studies varied from 4 months to 2 years. In the study by Işık et al. from 2021, implants were placed simultaneously with the GBR procedure. However, in other studies, the placement of implants varied from 3.5 months until 8 months after the sinus lift procedure.

In accordance with the objective, we defined the results at the outset as follows: histomorphometric outcomes and X-ray outcomes. For histomorphometric outcomes, one study did not perform any histological analysis [35]. Two studies [36,39] performed histomorphometric analysis 6 months after sinus lift, and the mean percentage of newly formed bone (NFB) varied from 12.95 to 21.25 in the control group (GC) and from 18.25 to 21.38 in the test group (GT). The mean percentage of residual graft (RG) varied from 28.54 to 32.72 in GC and from 19.16 to 25.95 in GT. No statistically significant difference was found between the two groups in these studies. One study [33] performed the analysis at 8 months after sinus lift, and the mean NFB, RG, and PST varied from 26.81 for GC to 28.69 for GT, 28.84 (GC) and 25.41 (GT), and 44.36 (GC) and 45.99 (GT), respectively. A statistically significant difference was found between GT and GC in terms of RG (*p* < 0.0001). No statistically significant difference was found between GT and GC in terms of NFB (*p* > 0.05) or PST (*p* > 0.5). Two studies [34,37] performed these analyses at 4 months, but in Pichotano’s study, the results were only reported for the GT. The NFB mean percentage varied from 39.49 in GC to 44.58 and 45.95 in GT, and the RG mean percentage varied from 15.62 in GC to 10.32 and 3.59 in GT. In the study of Irdem et al. from 2021, no statistically significant difference was observed between the two groups, and this evaluation was not possible in the study of Pichotano et al. from 2019, although they compared the two groups at different durations. The study in [38] performed histomorphometric analysis three times at 106, 120, and 150 days; NFB and RG were not measured, and no statistical analysis was performed. Despite the disparities in parameters and timing investigated in the different studies, there were no apparent differences among GC and NFB and GC.

As indicated above, Işık et al. (2021) did not perform histomorphometric analysis but did perform X-ray analysis, the results of which were interesting. The radiographic analysis by CBCT at 6 months indicated a variation in the mean percentage of residual bone width and an augmented thickness (AT) of 1.34, 2.49, and 2.97 mm according to the site (coronal, medial, and apical, respectively) in GC and 1.63, 2.59, and 3.11 in GT, respectively. Statistically significant differences between GT and GC were found in AT measured coronally, medially, and apically (*p* < 0.001, *p* = 0.007, and *p* = 0.036, respectively). This data indicate that PRF reduces bone resorption in mandibular guided bone regeneration with BioOss^TM^ in simultaneous implant placement. The implant survival rate was 100% in both groups, and healing was uneventful in all patients, with no significant pain or signs of infection in all studies.

### 3.3. Analysis of the Risk of Bias

The analysis of risk of bias as described in Figure 2 shows a low-to-moderate level of risk for all items in the different studies. The risk of selection, indication, attrition, and reporting bias is low overall due to the randomized nature of the included studies but also due to the lack of data loss during follow-up. However, it should be noted that the studies by Irdem et al., Isik et al. and Zhang et al. [34,35,39], have a high level of risk of bias for the item “blinding of participants and personnel”. This performance bias is explained by the lack of blinding of personnel and participants during surgery and biomaterial placement. In the studies by Irdem et al., Nizam et al., Pichotano et al. and Tatullo et al. and Zhang et al. [34,36,37,38,39], the risk of indication, performance, detection, and reporting data bias is moderate because the authors do not provide enough methodological information. However, the inclusion and exclusion criteria for participants remain similar across studies, ensuring homogeneity of the groups. The participants were their own controls in four studies (split-mouth design) or randomized in a 1:1 sequence (three studies). The comparability of data between the test and control groups was very good, and no bias was detected, except in the Pichotano study, where the histomorphometric analyses were performed at 4 months in the test group and at 8 months in the control group, which introduced bias to the comparison of results between the test and control groups.

## 4. Discussion

Recent systematic reviews evaluated the effectiveness of PRF or L-PRF alone or associated with different bone graft biomaterials in sinus floor augmentation [24,26,40]. In our review, we focused on the effectiveness of PRF or L-PRF associated with deproteinized bovine bone mineral in ridge augmentation procedures, including horizontal and/or vertical bone augmentation in the posterior maxillary or mandibular region. This systematic review included seven RCTs comparing the use of BioOss^TM^ alone or in combination with L-PRF, four of which had a split-mouth design. There were six RCTs on sinus lift surgery and one study on mandibular ROG. Overall, the studies had a mean follow-up time of 6 months, ranging from 4 months to 8 months, which would appear to be somewhat average for histomorphometric analysis. The number of subjects included varied from 7 to 60, with a general mean age of 51.20 years old. The studies showed considerable variation in the time taken to perform histomorphometric analysis. There were studies that performed the analysis at 8 months after bone augmentation surgery [26,33], 4 months [34,37], 6 months (Nizam et al. and Zhang et al.) [36,39], and 106 days [38]. Therefore, a meta-analysis is not applicable.

Globally, histomorphometric (new bone formation, percentage of residual bone, soft tissue, and mature bone) as well as radiological data (residual bone width and height, increased bone height, average graft volume, and increased thickness) show no statistically significant differences between the test group (L-PRF/PRF and DBBM) and the control group (DBBM alone) in the sinus lift surgical procedure. Our results corroborate those obtained in the meta-analyses by Canellas et al. from 2021 [41] and Damsaz et al. from 2020 [40]. These studies reported that the addition of PRF and L-PRF to BioOss^TM^ did not significantly improve the amount of regenerated bone after the sinus lift surgical procedure.

Ortega-Mejia et al. in their systematic review in 2020 [26] assessed the additional beneficial effects of platelet-rich fibrin (PRF) in combination with other bone grafting biomaterials. This review included 11 studies and concluded that there was no robust evidence regarding the beneficial effects of additional platelet concentrates on new bone formation in sinus augmentation. However, some studies included in the review reported favorable outcomes of PRF alone regarding implant survival, bone gain, and bone height, as well as improvement in the healing period and bone formation [42,43].

In our systematic review, we looked not only at sinus lift procedures but also at guided bone regeneration procedures. We found only one study that met the inclusion criteria [35]. The objective of the study was to assess augmentation success after guided bone regeneration (GBR) carried out simultaneously with implant placement using bovine-derived xenografts alone and in combination with liquid platelet-rich fibrin (liquid-PRF). A total of 20 patients with 50 implants were analyzed in the test group, and 20 patients with 48 implants were analyzed in the control group. The results show less bone resorption after grafting and implant placement, particularly in the group where L-PRF is added to DBBM. There are very few studies evaluating the benefits of adding PRF to graft materials in GBR.

There were few studies included in this review, and the parameters observed were variable and did not allow us to carry out a meta-analysis. Only one study combined PRF with BioOss in guided bone regeneration. In this systematic review, we included only RCTs, which are studies with high levels of scientific evidence (Oxford Levels of Evidence, 2011) For this procedure, more studies are needed to establish more relevant conclusions.

Concerning the new bone formation, the majority of the studies included in the systematic review did not show any real benefit to the use of L-PRF with BioOss^TM^. In a recent study from 2023, Dragonas et al. suggest that the addition of A-PRF and PRGF to DBBM does not enhance new bone formation outcomes in maxillary sinus augmentation procedures [44].

## 5. Conclusions

Overall, there were few studies, and the studies had many methodological differences. The analysis of the seven available articles concerning the new bone formation as primary outcome shows no real benefit to the use of L-PRF with BioOss^TM^. RCTs with similar methodologies are needed to confirm the available results and to provide recommendations for clinical practice.

## Figures and Tables

**Figure 1 jfb-14-00389-f001:**
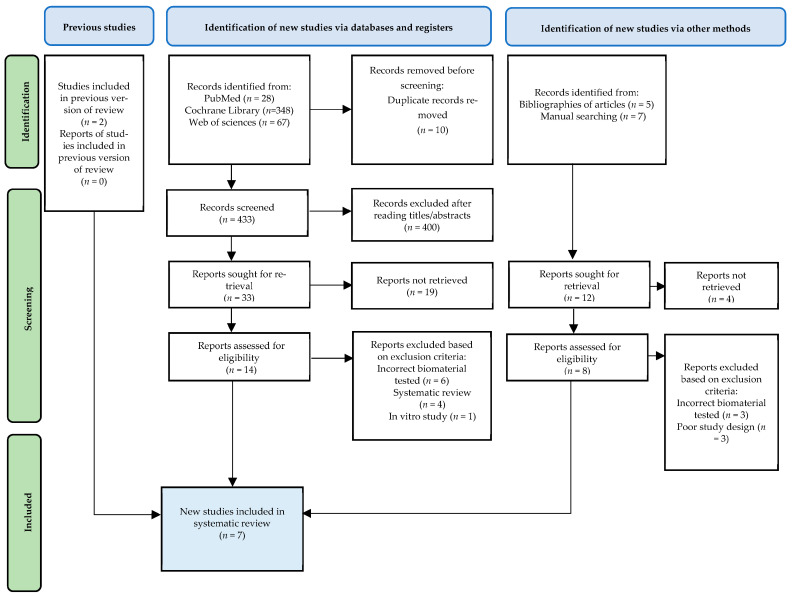
PRISMA flow diagram.

**Figure 2 jfb-14-00389-f002:**
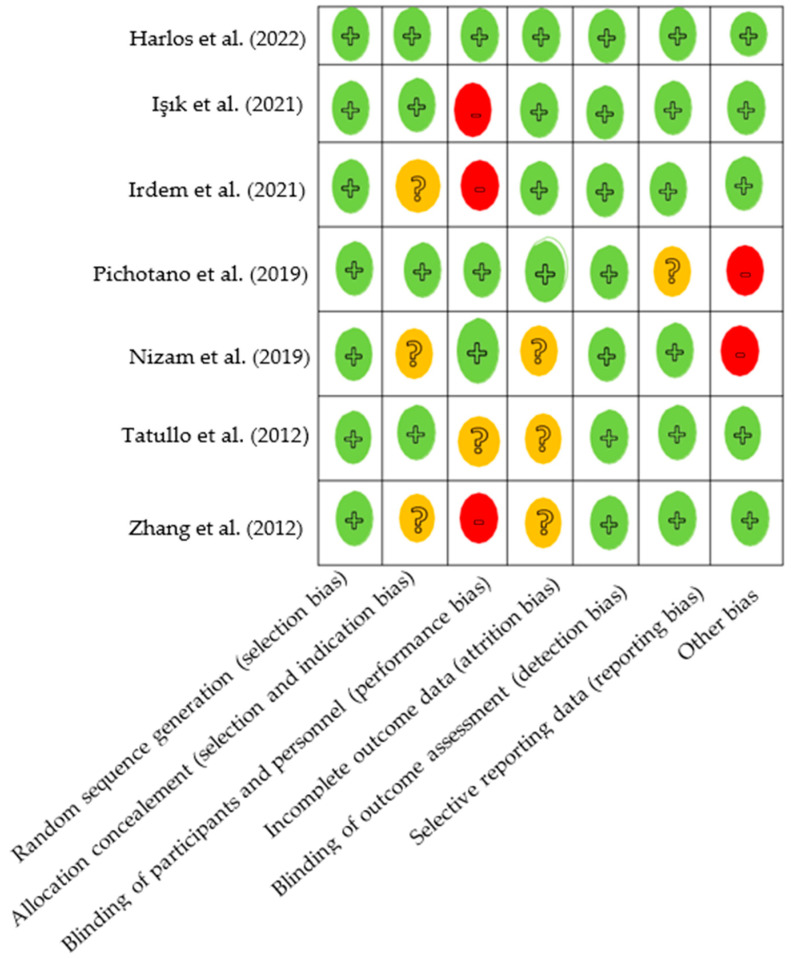
Risk of bias of included studies. Green indicates low risk of bias, orange indicates uncertain or moderate risk of bias, and red indicates high risk of bias [33,34,35,36,37,38,39].

**Table 1 jfb-14-00389-t001:** Databases and search terms.

**Pubmed (filters applied: Randomized Control Trial, Clinical trials)**	(“Bio-Oss” [Mesh] OR deproteinized bovine bone mineral OR bovine-derived xenograft OR bovine bone OR xenograft OR bone grafts OR bone substitutes) AND (“Platelet-Rich Fibrin” [Mesh] OR fibrin, platelet-rich OR platelet rich fibrin OR leukocyte- and platelet-rich fibrin OR leukocyte and platelet rich fibrin OR l-PRF OR PRF) AND (“Alveolar Ridge Augmentation” [Mesh] OR augmentation, alveolar ridge OR ridge augmentation, alveolar OR mandibular ridge augmentation OR maxillary ridge augmentation OR “Sinus Floor Augmentation” [Mesh] OR augmentation, sinus floor OR maxillary sinus augmentation OR sinus lifting OR maxillary sinus lift OR horizontal ridge augmentation OR vertical ridge augmentation)
**Cochrane library (All text)**	Bio-Oss OR deproteinized bovine bone mineral OR bovine-derived xenograft OR bovine bone OR xenograft OR bone grafts OR bone substitutes AND Platelet-Rich Fibrin OR fibrin, platelet rich OR platelet rich fibrin OR leukocyte and platelet rich fibrin OR leukocyte and platelet rich fibrin OR l-PRF OR PRF AND Alveolar Ridge Augmentation OR augmentation, alveolar ridge OR ridge augmentation, alveolar OR mandibular ridge augmentation OR maxillary ridge augmentation OR Sinus Floor Augmentation OR augmentation, sinus floor OR maxillary sinus augmentation OR sinus lifting OR maxillary sinus lift OR horizontal ridge augmentation OR vertical ridge augmentation
**Web of Sciences (All Fields; Articles)**	Bio-Oss AND Platelet-Rich Fibrin AND Alveolar Ridge Augmentation OR Sinus Floor Augmentation AND Human Clinical Trials

**Table 2 jfb-14-00389-t002:** Comparative table of studies using L-PRF and DBBM.

Author, Year of Publication, Country	Study Design	Nb. of Patients (Mean Age ± SD	Study Group	Nb. of Implants per Group (n) (Implant References)	Follow-up	X-ray Outcomes	Histomorphometric/Histological Outcomes	Findings
Harlos et al., 2022, Brazil [33]	RCT	36 (53.8 ± 4.6)	G1 (*n* = 12): DBBM (Bio Oss, Geistlich Pharma AG, Wolhusen, Switzerland) + autogenous bone G2 (GT) (*n* = 12): DBBM + L-PRF centrifuged at 2700 rpm for 12 min G3 (GC) (*n* = 12): DBBM alone Patients were treated for one maxillary sinus atrophy (sinus lift)	G1 (*n* = 24) G2 (*n* = 24) G3 (*n* = 24) Implants were placed 8 months after sinus lift (NS)	Clinical and radiographical follow-up at 8 months	(NS)	Histomorphometric evaluation was performed 8 months after sinus lift G1 NFB (39.97 ± 2.50%) RG (31.15 ± 3.39%) PST (28.88 ± 4.88%) G2 NFB (28.60 ± 2.32%) RG (25.41 ± 1.71%) PST (45.99 ± 2.71%) G3 NFB (26.81 ± 1.83%) RG (28.84 ± 3.55%) PST (44.36 ± 2.67%)	A statistically significant difference was found between GT and GCs in RG (*p* < 0.0001) No statistically significant difference was found between GT and GCs in NFB (*p* > 0.05) and PST (*p* > 0.5)
Işık et al., 2021, Turkey [35]	RCT	40 (49.99 ± 7.73)	GT (*n* = 22): DBBM (Bio Oss, Geistlich Pharma AG, Wolhusen, Switzerland) + L-PRF centrifuged at 700 rpm for 3 min + resorbable membrane (Bio-Gide^TM^, Geistlich Pharma AG, Wolhusen, Switzerland) GC (*n* = 22): DBBM + resorbable membrane Patient were treated for guided GBR in bone deficiency in the mandibular posterior region	GT (*n* = 50) GC (*n* = 48) Implants were placed simultaneously with the GBR procedure. Implant diameter was between 3.8 and 4.2 Ø and length was between 10 and 11 mm	Clinical and radiographical follow-up immediately after surgery and at 6 months, 1 year, and 2 years	CBCT after 6 months GT RBW (4.25 ± 0.26 mm) AT measured coronally (1.63 ± 0.21 mm) AT measured medially (2.59 ± 0.34 mm) AT measured apically (3.11 ± 0.36 mm) GC RBW (4.33 ± 0.28 mm) AT measured coronally (1.34 ± 0.14 mm) AT measured medially (2.49 ± 0.24 mm) AT measured apically (2.97 ± 0.24 mm)	(NS)	No statistically significant difference was found between groups in X-ray parameters before RGB in RBW (*p* = 0.512). A statically significant difference was observed between GT and GC in AT measured coronally, medially, and apically (*p* < 0.001, *p* = 0.007, and *p* = 0.036, respectively)
Irdem et al., 2021, Turkey [34]	RCT (split-mouth)	7 (50.57 ± 11.73)	GT (*n* = 7): DBBM (Bio Oss, Geistlich Pharma AG, Wolhusen, Switzerland) + L-PRF prepared from 9 mL blood sample tubes centrifuged at 2300 rpm for 15 min GC (*n* = 7): DBBM alone Patients were treated for bilateral maxillary sinus atrophy (sinus lift)	GT (*n* = 7) GC (*n* = 7) Implants were placed 4 months after sinus lift (NS)	Clinical and radiographical follow-up at 1 week, 1 month, 4 months, and 2 years	Panoramic X-rays after 4 months GT: RBH (3.77 mm) GC: RBH (3.88 mm)	Histomorphometric evaluation was performed 4 months after sinus lift GT: NFB (45.95%), MB (14.40%), RG (10.32%), FTR (29.31%), osteocalcin score (2.81 ± 0.36) GC: NFB (39.49%), MB (15.66%), RG (15.62%), FTR (28.59%), osteocalcin score (2.70 ± 0.39)	No statistically significant difference was found between groups in histomorphometric and X-ray parameters after sinus augmentation (*p* > 0.05). No problems were observed in any of the implants during the 2 year follow-up period.
Picotano et al., 2019, Brazil [37]	RCT (split-mouth)	12 (54.17 ± 6.95)	GT (*n* = 6): DBBM (Bio Oss, Geistlich Pharma AG, Wolhusen, Switzerland) + L-PRF centrifuged at 3000 rpm for 10 min + resorbable membrane (Bio-Gide^TM^, Geistlich Pharma AG, Wolhusen, Switzerland) GC (*n* = 6): DBBM + resorbable membrane Patients were treated for bilateral maxillary sinus atrophy (sinus lift)	GT (*n* = 19) GC (*n* = 19) Implants were placed 4 months after the sinus lift for the GT and 8 months after the sinus lift for the GC (NS)	Clinical and radiographical follow-up at 1 week and 4 months in the GT and 8 months in the GC	CBCT GT after 4 months MGV 1.10 ± 0.25 cm^3^ GC after 8 months MGV 0.91 ± 0.35 cm^3^	Histomorphometric evaluation was performed 4 months after sinus lift GT 4 months: NFB (44.58 ± 0.73%), RG (3.59 ± 4.22%), PST (26.60 ± 11.13%) GC 8 months: NFB (30.02 ± 8.42%), RG (13.75 ± 9.99%), PST (30.64 ± 12.46%)	No statistically significant difference was found in histomorphometric parameters after sinus augmentation in the GT (4 months) and GC (8 months) PST (*p =* 0.376) Statistically significant differences were found in histomorphometric parameters after sinus augmentation in the GT (4 months) and GC (8 months) NFB (*p* = 0.0087) RG (*p* = 0.011) No statistically significant difference was found in mean graft volume in GT and GC
Nizam et al., 2018, Turkey [36]	RCT (split-mouth)	13 (49.92 ± 10.37)	GT (*n* = 13): DBBM (Bio Oss, Geistlich Pharma AG, Wolhusen, Switzerland) + L-PRF centrifuged at 4000 rpm for 12 min GC (*n* = 13): DBBM alone Patients were treated for bilateral maxillary sinus atrophy (sinus lift)	GT (*n* = 30) GC (*n* = 28) Implants were placed 6 months after sinus lift Bone level implants [Institut Straumann AG, Basel, Switzerland (test: 11 and control: 11) and Zimmer TSV Implant System, Carlsbad, CA, USA (test: 19 and control: 17)] with diameters between 4.1 and 6.0 Ø and lengths between 10 and 13 mm	Clinical and radiographical follow-up at 6 months and 12 months	Panoramic X-rays after 6 months GT: RBH (2.45 ± 0.79 mm), ABH (13.60 ± 1.09 mm) GC: RBH (2.53 ± 0.61 mm), ABH (13.53 ± 1.20 mm)	Histomorphometric evaluation was performed 6 months after sinus lift GT: NFB (21.38 ± 8.78%), RG (25.95 ± 9.54%) RG in contact with NFB (47.33 ± 12.33%), PST (test; 52.67 ± 12.53%) GC: NFB (21.25 ± 5.59%), RG (32.79 ± 5.89%) RG in contact with NFB (54.04 ± 8.36%), PST (45.96 ± 8.36%)	No statistically significant difference was found between both groups in histomorphometric parameters after sinus augmentation: NFB (*p* = 0.96), RG (*p* = 0.06), RG in contact with NFB (*p* = 0.16) No statistically significant difference was found between groups in X-ray parameters after sinus augmentation No problems were observed in any of the implants during the 12 month follow-up period
Tatullo et al., 2012, Italy [38]	RCT (split mouth *n* = 12)	60 (55.15 ± NS)	GT (*n* = 36): DBBM (Bio Oss, Geistlich Pharma AG, Wolhusen, Switzerland) + PRF centrifuged 3000 rpm for 10 min GC (*n* = 24): DBBM alone 48 patients were treated for one maxillary sinus atrophy (sinus lift) and 12 were treated for bilateral maxillary sinus atrophy	Total (*n* = 120) GT = NS GC = NS	Radiographical follow-up at 6 months after implantation		Histomorphometric evaluation was performed 106 days (T1), 120 days (T2) and 150 (T3) days after sinus lift GT: TB (T1) 22.79% (T2) 26.15% (T3) 37.06% PST (T1) 70.2% (T2) 70.01 (T3) 61.41% GC: TB (T1) 26.44% (T2) 28.7% (T3) 38.97% PST (T1) 68.44% (T2) 68.18% (T3) 58.15%	No statistical analysis was performed
Zhang et al., 2012, China [39]	RCT	10 (44.85 ± NS)	GT (*n* = 5): DBBM (Bio Oss, Geistlich Pharma AG, Wolhusen, Switzerland) + PRF prepared at 3000 rpm for 13 min GC (*n* = 5): DBBM alone All patients were treated for unilateral maxillary sinus atrophy (sinus lift) except one patient treated for bilateral maxillary sinus atrophy in the test group	GT (*n* = 6) GC (*n* = 5) Implants were placed 6 months after the sinus lift (Replace Nobel Biocare)	Clinical follow-up at 1 week, 1 month, 3 months, and 6 months Radiographical follow-up immediately after the intervention and at 3 months and 6 months	Panoramic X-rays after 6 months (NS)	Histomorphometric evaluation was performed 6 months after sinus lift GT: NFB (18.35 ± 5.62%) RG (19.16 ± 6.89%) RG in contact with NFB (21.45 ± 14.57%) GC: NFB (12.95 ± 5.33%) RG (28.54 ± 12.01%) RG in contact with NFB (18.57 ± 5.39%)	No statistically significant difference was found between groups in histomorphometric parameters after sinus augmentation: NFB (*p* = 0.138), RG (*p* = 0.141), RG in contact with NFB (*p* > 0.05)

RCT, randomized controlled trial; Nb, number; GT, group test; GC, group control; L-PRF, liquid platelet-rich fibrin; DBBM, deproteinized bovine bone mineral; NS, not specified; GBR, guided bone regeneration; RBH, residual bone height; RBW, residual bone width; NFB, newly formed bone; MB, mature bone; RG, residual graft; FTR, fibrous tissue ratio; MGV, mean graft volume; ABH, augmented bone height; AT, augmented thickness; PST, percentage of soft tissue; TB, trabecular bone.

## Data Availability

Not applicable.

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
