# Peer review of "The Effectiveness of the Addition of Platelet-Rich Fibrin to Bovine Xenografts in Sinus and Bone Ridge Augmentation: A Systematic Review"

_jfb, 2023, doi:10.3390/jfb14070389_

Round 1

Reviewer 1 Report

The manuscript reported a systematic review to determine the effectiveness of the addition of platelet-rich fibrin to deproteinized bovine bone mineral in bone ridge augmentation. The number of included articles is 7, which is sufficient for a systematic review. Unfortunately, due to the significant heterogeneity in the included studies, meta-analysis is not performed. 

The search strategy, study screening and selection, and data extraction were performed systematically and scientifically. The data reporting and discussion were also adequately done. Thus, the manuscript can be considered for publication in the current form.

Minor comments:

1. Use author and year, instead of the reference number when starting a sentence. E.g., at lines 199 and 226.

Author Response

Dear reviewer,  
We would like to thank you for the time you spent  reviewing  our artcile and for your pertinent remarks.
We included in the text the modifications you asked.
Once again, thank you very much for your work.

Reviewer 2 Report

 The study is a systemic review to evaluate the adjunction of PRF or L-PRF to DBBM in bone ridge augmentation procedures. The article is interesting to clarify the effect of platelet-rich fibrin in ridge augmentation. From the study’s objectives (p3, line 82-85) stated that the study  aimed to evaluate the effect of PRF when mixed with BioOssTM in comparison with the use of BioOssTM alone in alveolar ridge augmentation and after bone augmentation procedures, the primary outcome was new bone formation evaluated with histomorphometric analysis. The secondary outcomes were other histomorphometric parameters, such as the percentage of residual bone and soft tissue, bone maturation and X-ray outcomes (P3, line 98-100).

However, there is only one study performed ridge augmentation and presented only secondary outcome of CBCT without histomorphometry (Ref 35). Others studies dealt with sinus augmentation, therefore the title and discussion should be rewritten to be “ in sinus augmentation” not in ridge augmentation. It should be noted that the effect of PRF in sinus augmentation is not new.

The other point, there is confusion regarding the term used in this paper, L-PRF normally referred to Leucocyte-platelet-rich fibrin which is the same as PRF, the authors used the term L-PRF for liquid PRF which is normally used as I-PRF or injectable PRF or liquid form. In the searching protocol, L-PRF has been used, so it referred to leucocyte-PRF. I don’t know that the mixture of DBBB with PRF using the liquid form to make a sticky bone or the fibrin form to mix. There are 4 articles gave positive effects from slight to significant effect, 1 gave equal effect, and the other one gave negative effect. The authors should analyze and draw discussion and conclusion. The article should be revised and rewritten meticulously, Ref 35 should be excluded. After revised it should be submitted in a new form.

Author Response

Dear reviewer 
We would like to thank you for the time you spent reviewing our artcile and for your  very pertinents remarks.

Concerning your first remark, we agree that the majority of the studies concern the sinus augmentation. In consequence we modified the title of the article and  the discsussion.

About the second remark, 5 of  the  7 studies included in the systematic review, used the L PRF form. The two others studies  are unclear but depending of the protocol of preparation, it seems to be also L-PRF. In all studies, the L-PRF is mixed with DBBM.

Concerning the primary outcome which is new bone formation, only one study (Picotano et al. 2019) showed a statistically significant difference between the control and the test groups.  It was clearly expressed in the discusion. We modified the conclusion. 

Concerning the reference N°35 we agree that this study doesn't use histomorphometric analysis and this limitation is clearly expressed in the text and the table 2. However, this article is the only one concerning the ridge augmentation in the mandible. In consequence it seemed to be important to include it in the systematic review.

Reviewer 3 Report

I appreciate the opportunity to review the manuscript for publication in MDPI JFB. I feel that the topics are interesting, and the manuscript is grossly organized. I have a few comments as follows.

 The systematic review aimed to evaluate the effect of PRF when mixed with BioOssTM in comparison with the use of BioOssTM alone in alveolar ridge augmentation and in the outcome of implantation after bone augmentation procedures.

The primary outcome was new bone formation evaluated with histomorphometric analysis. The secondary outcomes were other histomorphometric parameters, such as the percentage of residual bone and soft tissue, bone maturation and X-ray outcomes.

Table 1 is difficult to see in ways of horizontal alignment.

L159: “17 records were excluded for reasons such as evaluation of another biomaterial than DBBM (n = 9), in vitro studies (n = 1), poor study design (n = 3), and systematic review (n = 4)”

I am curious about results from other biomaterials. Are there any data showing superiority to the DBBM?

The literature order in Table 2 and Figure 2 should be in same manners.

The authors conclude that the analysis of the 7 available articles shows no real benefit to the use of PRF with Bio-OssTM. RCTs with similar methodologies are needed to confirm the available results and to provide recommendations for clinical practice.

The procedure of meta-analysis should be employed to ascertain this vital question.

Author Response

Dear Reviewer

We would like to thank you for the time you spent reviewing our article and for your pertinent remarks.

We included in the text the modifications you asked.

Once again, thank you very much for you work.

Concerning your first remark, DBBM is the most used and investigated biomaterial in the literature and represents the gold standard in pre-implant surgery. Concerning the others biomaterials, only very limited data are available for each biomaterial and it’s impossible to conclude to any superiority.

Concerning the second remark, Table 2 was modified and the literature order is now similar in table 2 and figure 2.

Corncrnig the third remark, there were few studies included in this review, and the parameters observed were variable and did not allow us to carry out a meta-analysis. 

Reviewer 4 Report

The authors of the article submitted for review are Katia Idiri, Octave Bandiaky, Assem Soueidan, Christian Verner, Emmanuelle Renard and Xavier Struilou  dental specialists working at the Faculté de Chirurgie Dentaire (Dentistry Department) in Nantes, France. They are part of a medical team operating at the Nantes University HospitalFaculté de Chirurgie Dentaire in Nantes is a well-known and respected academic center participating in dental education, scientific research as well as providing treatment for patients. Dentists, such as the authors of this manuscript, play a critical role in conducting research on the subject of dentistry whilst simultaneously providing their patients with complex dental care. 

The article I was presented with is a systematic review of literature trying to assess the effectiveness of the addition of platelet-rich fibrin (PRF) or liquid platelet-rich fibrin (L-PRF) to bovine xenographs with DBBM (deproteinized bovine bone mineral) used in bone ridge augmentation procedures. The authors have conducted a systematic review of various randomized clinical research, where either PRF or L-PRF were combined with bovine DBBM and used in jaw bone augmentation

Currently, the utilization oPRF in dental or maxillofacial surgery, is becoming more and more popular and therefore thismaterial is now widely-spread also in bone augmentationproceduresThe already existing research and knowledge on the subject of DBBM has been well-established and supported by evidence. What still requires additional studies is whether the addition of PRF to DBBM indeed improves the treatmentoutcome. Therefore, I believe that the topic chosen for this review is very up to date and interesting which serves to the authorsbenefit. 

After conducting the initial literature analysis, taking into account all of the inclusion and exclusion criteria, only 7 clinical papers qualified to be evaluated and incorporated into this article. Unfortunately, for that reason, as well as the fact that the observed parameters were variable, no data meta-analysis was performed.Those circumstances significantly decrease the scientific value of the manuscript. The results that were obtained by the authors show that the addition of PRF to DBBM in comparison with using just DBBM (no PRF added) in a sinus lift procedure has no statistically beneficial properties. The article also discusses existing research that proves the appropriateness of implementing PRF in other dental procedures such as soft tissue regeneration, periodontology or maxillary sinus floor augmentationThe article also presents studies suggesting that the use of PRF in those procedures resulted in very positive outcomeIn my personal opinion, such digressions are not applicable to a systematic review of RCTs and should be removed. 

The article exhibits all of the collected data in the form of tables and chartsThey are very clear and logical which makes them easy and quick to analyse by the reader. I also have no major reservations about the linguistic aspect of this manuscript. It is well-written and understandable despite the use of professional terms and academic language and the authors being non-native speakers of English

Among a few critical comments and questions that I have regarding this work, that obviously need to be addressed by the authors, I would like to especially focus on these two

Because the topic of this manuscript is a very popular one, and the number of publications selected for this analysis is rather small in comparison, it would be beneficial to expand the area of research by adding a scoping review  according to the Oxford Centre for EMB 2011This type of review, in my eyes, would be way more useful for the JFB readers and would meet the expectations that they have towards this journal.

- I suggest adding at least a few publications from the current year 2023. 

Author Response

Dear Reviewer

We would like to thank you for the time you spent reviewing our article and for your pertinent remarks.

We included in the text the modifications you asked.

Once again, thank you very much for you work.

Concerning your first remark, we added in the discussion a sentence concerning the level of scientific evidence of the studies included in the systematic review.

Concerning your second remark, we performed an electronic research and added a recent 2023’s reference in the discussion.

Round 2

Reviewer 3 Report

I appreciate the opportunity to review again the manuscript for publication in MDPI JFB. I reckon that the manuscript has been revised and improved in part in accordance with the reviewers’ comments.

Author Response

Dear reviewer

we would like to thank you for your remarks to improve our paper and to have accepted our article

Reviewer 4 Report

Dear Authors,

In my review I pointed out that one of the most important aspects of the manuscript that needs to be addressed is changing it to a scoping review which is a preliminary assessment of potential size and scope of available research literature that identifies the nature and extent of research evidence and oftentimes includes ongoing research. There was no explanation in your reply as to why this was not done. Simply adding one sentence in the “Discussion” section does not meet the requirements of a scoping review. The authors also did not relate to my comment that there is no place for presenting not yet scientifically proven studies on potential PRF addition benefits in those procedures in a systematic review and it was not removed. 

Therefore, the text still needs major revisions as the only change that was made to it in accordance with my criticism was updating the literature with publications from 2023.

Author Response

dear reviewer,

Once again, thank you very much to have spent time to revise our article.

I'm really sorry but i do not accept to change the article from a systematic review to a scopic review. This proposition of systematic review was registered and accepted on Prospero and we have done this work in respect with teh specific procedures needed in a systematic review. more, there was many systematic review on the use of PRF in Dentistry,published in the literature during the last years. these publications were done in serious reviews as Clinical Oral Implant Research, Clinical Oral Investifgation, Materials or Journal of Periodontology as examples. That is the reason why we decided to keep this article as a systematic review but we integrated your remark in the discussion adressing the scientific level of evidence of the publications included in this systematic review. Moreover, the principle of a systematic review on this topic has been accepted by all the other reviewers. All there suggestions have been included and accepted in the version you revised.

We would like to thank you again for your precious time and your remarks

Kind regards